Comparing local ancestry inference models in populations of two- and three-way admixture

Schubert Ryan 1 2 3
Andaleon Angela 2 3
Wheeler Heather E. hwheeler1@luc.edu 2 3 4
1 Department of Mathematics and Statistics, Loyola University Chicago , Chicago , IL , United States of America
2 Department of Biology, Loyola University Chicago , Chicago , IL , United States of America
3 Program in Bioinformatics, Loyola University Chicago , Chicago , IL , United States of America
4 Department of Public Health Sciences, Loyola University Chicago , Maywood , IL , United States of America
Ge Jianye
Electronic publication date: 2020 Oct 2
Publication date: 2020
Volume: 8
Electronic Location ID: e10090
Received 2020 Jun 25; Accepted 2020 Sep 13
Copyright: ©2020 Schubert et al.
Copyright year: 2020
Copyright holder: Schubert et al.
License: This is an open access article distributed under the terms of the Creative Commons Attribution License, which permits unrestricted use, distribution, reproduction and adaptation in any medium and for any purpose provided that it is properly attributed. For attribution, the original author(s), title, publication source (PeerJ) and either DOI or URL of the article must be cited.
License URL: https://creativecommons.org/licenses/by/4.0/

Keywords: Local ancestry, Human genetics, Admixture, Benchmarking, Population genetics

Funding: National Institutes of Health National Human Genome Research Institute Academic Research Enhancement Award R15 HG009569 Loyola University Chicago Carbon Undergraduate Research Fellowship Loyola MS Bioinformatics Research Assistant Fellowship This work is supported by the National Institutes of Health National Human Genome Research Institute Academic Research Enhancement Award R15 HG009569 (PI: Heather E. Wheeler), the Loyola University Chicago Carbon Undergraduate Research Fellowship (Angela Andaleon), and the Loyola MS Bioinformatics Research Assistant Fellowship (Angela Andaleon). The funders had no role in study design, data collection and analysis, decision to publish, or preparation of the manuscript.

==============================
Local ancestry estimation infers the regional ancestral origin of chromosomal segments in admixed populations using reference populations and a variety of statistical models. Integrating local ancestry into complex trait genetics has the potential to increase detection of genetic associations and improve genetic prediction models in understudied admixed populations, including African Americans and Hispanics. Five methods for local ancestry estimation that have been used in human complex trait genetics are LAMP-LD (2012), RFMix (2013), ELAI (2014), Loter (2018), and MOSAIC (2019). As users rather than developers, we sought to perform direct comparisons of accuracy, runtime, memory usage, and usability of these software tools to determine which is best for incorporation into association study pipelines. We find that in the majority of cases RFMix has the highest median accuracy with the ranking of the remaining software dependent on the ancestral architecture of the population tested. Additionally, we estimate the O(n) of both memory and runtime for each software and find that for both time and memory most software increase linearly with respect to sample size. The only exception is RFMix, which increases quadratically with respect to runtime and linearly with respect to memory. Effective local ancestry estimation tools are necessary to increase diversity and prevent population disparities in human genetics studies. RFMix performs the best across methods, however, depending on application, other methods perform just as well with the benefit of shorter runtimes. Scripts used to format data, run software, and estimate accuracy can be found at https://github.com/WheelerLab/LAI_benchmarking.

Introduction

Humans are a chromosomal mosaic of their ancestors. Through sexual reproduction and recombination, chromosomes resemble a subset of their ancestors’ chromosomes in varying sizes and locations across the genome (Gravel, 2012). Large scale studies of the genetics underlying human disease have been limited to predominantly European populations and thus lack global diversity, which exacerbates health disparities (Popejoy & Fullerton, 2016; Sirugo, Williams & Tishkoff, 2019). It is well documented that prediction accuracy with polygenic risk scores decreases with increasing genetic distance (Martin et al., 2017; Mogil et al, 2018). In addition, many underrepresented populations in human genetics include recently admixed individuals, meaning their ancestors were previously isolated from each other on different continents until the last few centuries. This leads to chromosomal tracts originating from different continental populations in modern populations like African Americans and Hispanics.

Population structure is a potential confounding factor in all genetic association studies. Global ancestry is the proportion of different ancestral populations represented across the entire genome. Genotypic principal components are used to adjust for these average genomic background effects in genetic association studies (Price et al., 2006). Correcting only for global ancestry does not precisely account for ancestry at any specific locus. Local ancestry is the number of alleles derived from distinct ancestral populations at a given locus and may improve power to detect genetic associations in admixed populations (Duan et al., 2017; Martin et al., 2017b; Zhang & Stram, 2014; Liu et al., 2013. For example, a recent expression quantitative trait locus (eQTL) mapping study in African Americans found a greater replication rate of eQTLs discovered via models that adjust for local ancestry, rather than models that adjust for global ancestry (Zhong, Perera & Gamazon, 2019).

Several models have been developed to estimate local ancestry in admixed populations (Gravel, 2012; Baran et al., 2012; Maples et al., 2013; Guan, 2014; Thornton & Bermejo, 2014; Dias-Alves, Mairal & Blum, 2018; Salter-Townshend & Myers, 2019). By leveraging population or continental-specific SNPs, chromosomal tracts can be differentiated into their ancestral segments. Chromosomal regions are compared to reference populations of less recently admixed ancestry to find which sections of the chromosomes descend from which continental region (Gravel, 2012; Bryc et al., 2015). These estimates depend largely on the reference populations used, the genetic distance between the reference samples, the quality of the input genotypes, and, most importantly, the statistical models. LAMP-LD demonstrates strong ancestry estimation in recently admixed cohorts of African and Hispanic descent (Baran et al., 2012). ELAI and Loter both report stable performance in instances of ancient admixture (ngenerations ≥ 100), outcompeting methods that prioritize recent admixture (Guan, 2014; Dias-Alves, Mairal & Blum, 2018). Additionally, Loter reports high performance in nonhuman species (Dias-Alves, Mairal & Blum, 2018). Similar to LAMP-LD, RFMix and MOSAIC each specialize in multi-way admixture. Unlike LAMP-LD, neither are constrained in the number of ancestral populations. Both RFMix and MOSAIC are reported to have robust performance even when reference panels are not closely related to the study population, though MOSAIC reported the added benefit of elucidating the relationship between all provided references and the study population in selecting the optimal references, thus circumventing the need to clarify the relationship between study and available reference populations (Maples et al., 2013; Salter-Townshend & Myers, 2019).

To satisfy the need for increased diversity in genome-wide association studies (Popejoy & Fullerton, 2016;Sirugo, Williams & Tishkoff, 2019), local ancestry estimation methods will become increasingly important in human genetics. While a recent review compared the underlying models of several local ancestry estimation software tools (Geza et al., 2018), accuracy and run time were not directly compared. A study from 2017 compared run time and memory usage of four older tools (Hui et al., 2017), but did not include the widely used RFMix (Maples et al., 2013) and newer tools MOSAIC (Salter-Townshend & Myers, 2019) and Loter (Dias-Alves, Mairal & Blum, 2018). Here, we independently compare five local ancestry estimation methods for accuracy and feasibility by simulating admixed chromosomes from both two and three ancestral continental populations.

Methods

Simulating genotypes

Our workflow is summarized in Fig. 1. We note that all populations are admixed at some point in their histories. In this paper, we define non-admixed populations as those whose individuals cluster tightly in multi-population principal component analysis and are thus typically used as ancestral reference populations in local ancestry studies. We chose three 1000 Genomes (1000G) populations (Auton et al., 2015) to serve as non-admixed ancestral populations. From each of these populations we randomly selected 10% of individuals to use as founders for simulation of admixed individuals and the remaining individuals made up the non-admixed reference populations. The three 1000G populations from which we drew samples are: Utah residents with Northern and Western European ancestries (CEU) for use as an European ancestral group; Yoruba in Ibadan, Nigeria (YRI) for use as an African ancestral group; and Peruvians from Lima, Peru (PEL) for use as a Native American ancestral group. We note that individuals in the PEL population have Native American, European and African admixture, however, the PEL have more Native American ancestries than all of the other American populations in 1000G (μ = 0.77, 95% CI [0.75–0.80] (Martin et al., 2017), where µis the estimated mean proportion of Native American ancestries). We restricted this analysis to individuals in the PEL population with Native American ancestry proportions greater than µto serve as a reasonable proxy for a Native American ancestral population. Simulated admixed populations fall into one of three categories: two-way admixture between YRI and CEU representing a common pattern of descent for African American individuals (AFA); two-way admixture between PEL and CEU representing one common pattern of descent for some Hispanic individuals (HIS); and three-way admixture between PEL, YRI, and CEU, representing another common pattern of descent among some Hispanic individuals (3WAY) (Bryc et al., 2015; Martin et al., 2017). For each admixture group, we simulated 1,000 individuals and selected 100 that had European ancestries within 10% of the admixture proportions described in Table 1 based on empirical proportions estimated in Bryc et al. (2015) and Martin et al. (2017a). Global ancestry percentages across individual haplotypes are shown in Figs.  S1–S3.

Figure 1 Process for simulating admixed individuals and estimating ancestries.

(1) From non-admixed populations from 1000G we randomly select 10% of all individuals to use as founders for admixture simulation. The rest are used as reference panels for ancestry estimation. (2) We generate admixed individuals that are chromosomal mosaics of the founder group using the admixture simulation tool created by the authors of RFMix (Maples et al., 2013). (3) Using the remaining 1000G populations as reference panels, we estimate ancestries on the simulated population for all five software tools and compare estimation accuracy.

Table 1 Proportions of ancestry among simulated cohorts.

We simulated 1,000 admixed individuals. From these we selected 100 individuals with a true proportion of European (CEU) ancestries within 10% of the proportions listed here for use in accuracy testing.

Ancestral panel	%YRI	%CEU	%PEL	
AFA	80%	20%	0%	
HIS	0%	70%	30%	
3WAY	30%	60%	10%	

We used the admixture simulation tool developed by the creators of RFMix to generate simulated admixed chromosomes (Maples et al., 2013). Because lengths and recombination rates differ among human chromosomes (Farr, Micheletti & Ruiz-Herrera, 2012), we chose to simulate admixture using a portion of SNPs on chromosome 1 and all SNPs on chromosome 22. We used the 44,332 SNPs in common among the reference populations within the first 10Mb of chromosome 1 and the 158,159 common SNPs on chromosome 22. LAMP-LD v 1.0 has a computational limit of 50,000 SNPs. In keeping with this, after simulating the entirety of chromosome 22, we independently selected 50,000 SNPs from each cohort using the –thin-count 50000 option in PLINK (Chang et al., 2015) and subset each cohort accordingly. In an effort to remove the noise that is introduced by admixed PEL individuals, we limit this population to only those individuals with a proportion of Native American ancestry that is above the mean estimated by Martin et al. (2017a) (μNAT = 0.77). The code used to run simulation can be found at https://github.com/WheelerLab/LAI_benchmarking.

Running each software

We used individuals remaining within the non-admixed ancestral group after founder selection as the required reference group for running each of the five software. We ran each software using default parameters or using the minimum number of settings necessary as this is representative of how most new users will interact with each software. We ran each software as follows:

LAMP-LD v1.0

unolanc  300 15 <snp  position  file> <ancestral  haploypes 1> <ancestral  haploypes 2> <ancestral  haploypes 3> <admixed haploypes> <output  name> unolanc2way  300 15 300 15 <snp  position  file>   <ancestral haploypes 1>   <ancestral  haploypes 2>   <admixed  haploypes> <output  name>

MOSAIC v1.3

Rscript  mosaic.R <admixed  population  name> <folder containing  required  input> −c <chr range> −a <number  of ancestries  to infer> −m <maximum  number  of cores> −−gens <number  of  generations>

Loter

loter_cli −r <reference  panel  genotype/haplotype> −a <admixed  genotype/haplotype> −f <genotype  file  format> −o <output  name> −n <number  of cores> −v

ELAI v1.01

elai−lin −g <ancestral  haploypes 1> −p 10 −g <ancestral haploypes 2> −p 11 −g <ancestral  haploypes 3> −p 12 −g <admixed  haploypes> −p 1 −pos <snp  position  file> −C 3 −o <output  name> elai−lin −g <ancestral  haploypes 1> −p 10 −g <ancestral haploypes 2> −p 11 −g <admixed  haploypes> −p 1 −pos <snp  position  file> −C 2 −o <output  name>

RFMix v1.5.4

python  RunRFMix.py −e 2 −w 0.2 −−num−threads <maximum number  of cores>   −−forward−backward  PopPhased <population  haploypes> <population  classes  file> <snp position  file> −o <output  name>

In all cases we ran software on one core. In cases with three ancestries, 11 was used for number of generations. In cases with two ancestries, 8 was used for number of generations. In most cases each software requires a genetic map file or SNP position file, the number of generations since admixture, and reference/admixed genotypes in a software specific format. As this genotype data was already phased, we do not consider phasing in this paper, though it could be considered a necessary step 0 of this process. As each software carries different requirements for formatting, we have constructed a pipeline for formatting and running each software. All scripts used to run each software can be found at https://github.com/WheelerLab/LAI_benchmarking.

Benchmarking each software

We used the bash command time -v to benchmark time and memory of each software run. To benchmark time and memory usage with increasing sample size, we used the methods described above and simulated an additional 2000 two-way admixed AFA individuals to test time and memory burden at each level of 20, 50, 100, 500, 1000, 1500, and 2000 individuals. We performed regression analysis of time and memory complexity in base R for each software.

We compared the estimated ancestries to the simulated (‘true’) ancestries to determine software accuracy. For each individual we calculated the true positive call rate (TPCR), which is the proportion of correct ancestry calls at each locus (SNP) in each individual. We then tested for differences in TPCR between software methods in each simulated population via Tukey’s test.

Results

We prioritize benchmarking each software in the context of recently admixed populations to assess accuracy and estimate previously unreported time and memory complexity. We selected five software tools for a combination of their novelty and relative popularity. LAMP-LD (Baran et al., 2012), ELAI (Guan, 2014), and RFMix (Maples et al., 2013) are each established local ancestry software that have been cited numerous times in the field of population genetics. Conversely, MOSAIC (Salter-Townshend & Myers, 2019) and Loter (Dias-Alves, Mairal & Blum, 2018) are fairly new, having been published in the last two years at the time of writing. A brief summary of their differences can be found in Table 2.

Table 2 Software descriptions.

Features and requirements of each software as described in the original publication.

Software	Algorithm	Pre-phasing	Gen map	n ancestral pops	
LAMP-LD	single layer HMM	not required	no	2,3,5	
RFMix	random forest	required	yes	n ≥ 2	
ELAI	two layer HMM	not required	no	n ≥ 2	
Loter	single layer HMM	required	no	n ≥ 2	
MOSAIC	two layer HMM	required	yes	n ≥ 2	
Notes.

HMM, hidden Markov model.

Simulating admixed individuals

We simulated admixed populations with ancestry proportions similar to those observed in previous studies (Bryc et al., 2015; Martin et al., 2017). These include two-way admixture between YRI and CEU representing a common pattern of descent for African American individuals (AFA); two-way admixture between PEL and CEU representing one common pattern of descent for some Hispanic individuals (HIS); and three-way admixture between PEL, YRI, and CEU, representing another common pattern of descent among some Hispanic individuals (3WAY) (Bryc et al., 2015; Martin et al., 2017). For each admixture group, we simulated 1000 individuals and selected 100 that had European ancestries within 10% of the admixture proportions listed in Table 1. We summarize this workflow in Fig. 1, see Methods for details.

Runtime and memory usage

Runtime increases with number of individuals

We simulated an additional 2000 individuals based on the AFA admixture proportions at 7 generations since admixture. We randomly subset this set of people to 2,000, 1,500, 1,000, 500, 100, 50, and 20 individuals to test how each software scales with an increasing sample size (Fig. 2). We find that the runtimes of four of the five software tools scale linearly with the number of samples, with the exception of RFMix, which scales quadratically (Table 3). We also note that MOSAIC runtime decreases when n = 2000. MOSAIC will exit early the iteration of its expectation-likelihood algorithm when the log-likelihood decreases resulting in in cases where it finishes faster than would be expected by a standard linear model (Salter-Townshend & Myers, 2019).

Figure 2 Software runtime versus sample size.

We tested the runtime of each software on one core at a sample sizes of 20, 50, 100, 500, 1,000, 1,500, and 2,000. Points represented sample sizes tested versus runtime in hours, which are connected by line segments colored by software. We simulated n African American individuals from CEU and YRI “founder” populations with average admixture proportions of 20% and 80%, respectively. We find that the runtime of ELAI, LAMP-LD, MOSAIC, and Loter all increase linearly with the number of samples. The runtime of RFMix increases quadratically.

Memory increases linearly with number of individuals

We simultaneously measured the memory burden expected for each level of sample size (Fig. 3). We found that all software expand linearly or near linearly (Table 4). Loter had the steepest memory requirement and ELAI had the smallest slope. ELAI has the most stable memory requirement across sample sizes. At high sample sizes ELAI had the lowest memory overhead, but at low sample sizes (n ≤ 100) the memory requirement was third highest.

Increasing number of ancestries can increase runtime and memory burden

We tested if increasing the number of ancestral populations increases the computational burden of each software. We found that increasing the number of ancestral populations increases the memory usage in ELAI and LAMPLD (Fig. 4) and the runtime in RFMix and ELAI (Fig. 5). For the remaining software, increasing the number of ancestries did not significantly increase the memory usage or runtime.

Table 3 Linear runtime estimated O(n).

We fit linear and quadratic models between the runtime and sample size for each software. We report the model R2 and ANOVA p-value for each combination of software and model.

Software	Linear R2	Linear p-value	Quadratic R2	Quadratic p-value	
RFMix	0.853	0.00294	0.977	1.57 × 10−05	
MOSAIC	0.673	0.0146	0.387	0.0802	
ELAI	0.999	8.97 × 10−11	0.921	0.000383	
Loter	0.999	2.62 × 10−13	0.91	0.000438	
LAMP-LD	0.999	7.90 × 10−14	0.917	0.000427	

Figure 3 Software memory usage versus sample size.

We tested the maximum memory usage of each software on one core at a sample size of 20, 50, 100, 500, 1,000, 1,500, and 2,000. Points represented sample sizes tested versus memory in gigabytes (GB), which are connected by line segments colored by software. We simulated n African American individuals from CEU and YRI “founder” populations with average admixture proportions of 20% and 80%, respectively. We find that maximum memory usage for all software increases linearly with the number of samples.

Table 4 Linear maximum memory usage estimated O(n).

We fit a linear model between the maximum memory usage and sample size for each software. We report the model R2 and ANOVA p-value for each combination of software and model.

Software	Linear R2	Linear p-value	Quadratic R2	Quadtratic p-value	
RFMix	0.999	8.61 × 10−12	0.9205	0.000394	
MOSAIC	0.997	6.049 × 10−08	0.894	0.000813	
ELAI	0.999	1.03 × 10−11	0.9154	0.000459	
Loter	1	1.25 × 10−14	0.9154	0.000459	
LAMP-LD	1	<2.2 × 10−16	0.9165	0.000445	

Figure 4 Software memory usage versus number of ancestral populations.

Comparison between memory usage in megabytes (MB) of each software and the number of ancestries estimated in 100 individuals simulated per chromosome in each population. Memory usage differed by number of ancestral populations using ELAI and LAMP-LD. One-way ANOVA results for each software: (A) ELAI p = 3.78 × 10−4, (B) LAMP-LD p = 3.67 × 10−2, (C) Loter p = 0.637, (D) MOSAIC p = 0.65, (E) RFMix p = 0.204.

Accuracy varies by cohort composition

For each admixture group (Table 1), we simulated 100 individuals and ran local ancestry estimation and accuracy benchmarking. Each software performs with high fidelity in the majority of instances but we note a noticeable difference in the performances of the AFA and HIS cohorts. We attribute this to the introduction of the PEL population as both founders and reference, as they contain a significant amount of admixture in and of themselves. As PEL admixture overlaps with the other two reference populations, it is expected that they will introduce noise into local ancestry estimation. In all instances RFMix had the highest median TPCR (Fig. 6). We pooled the TPCRs between chromosomes per software to test which pairs of software performed differently in each simulated population using Tukey’s test. In the case of the simulated AFA cohort, both LAMP-LD and RFMix performed significantly better than each of ELAI, MOSAIC, and Loter and both ELAI and Loter performed significantly better than MOSAIC. All other pairs were not significantly different (Fig. S4). In the case of the HIS cohort, RFMix performed significantly better than all other software; ELAI, MOSAIC, and LAMP-LD performed significantly better than Loter; and ELAI performed significantly better than MOSAIC (Fig. S5). In the case of the 3WAY cohort, RFMix performed significantly better than all other software, LAMP-LD and ELAI were both significantly better than both Loter and MOSAIC, and MOSAIC performed significantly better than Loter (Fig. S6). In the majority of cases for both chromosome 1 and chromosome 22, RFMix demonstrates the highest true positive rate for each population tested (Fig. 6).

Figure 5 Software runtime versus number of ancestral populations.

Comparison between runtime in seconds (s) of each software and the number of ancestries estimated in 100 individuals simulated per chromosome in each population. Runtimes differed by number of ancestral populations using ELAI and RFMix. One-way ANOVA results for each software: (A) ELAI p = 5.98 × 10−3, (B) LAMP-LD p = 0.906, (C) Loter p = 0.985, (D) MOSAIC p = 0.555, (E) RFMix p = 2.01 × 10−3.

Figure 6 Distribution of accuracy for ancestry estimation.

For each category of admixture, (A) 3WAY, (B) AFA, (C) HIS, we estimate ancestries on 100 simulated individuals. Accuracy is then calculated as the true positive call rate for the estimated ancestries per each software. True positive call rate (TPCR) is defined as the proportion of loci in each individual with an ancestry correctly estimated by a given software. RFMix had the highest TPCR across most pairwise comparisons. Tukey’s test results are presented in Figs. S4–S6.

Table 5 Between software Pearson correlation using real data.

We ran all five software on 61 real admixed individuals from the 1000 Genomes ASW (African Ancestry in Southwest US) population. Here we report the squared pairwise Pearson correlations of local ancestry estimates. Additionally, in the last column, we report the squared correlation of each software’s estimated mean African ancestry with genotypic principal component 1.

	ELAI	LAMP-LD	Loter	RFMix	MOSAIC	PC1	
ELAI	1	–	–	–	–	0.965	
LAMP-LD	0.977	1	–	–	–	0.968	
Loter	0.976	0.981	1	–	–	0.967	
RFMix	0.974	0.977	0.977	1	–	0.967	
MOSAIC	0.959	0.959	0.962	0.961	1	0.960	

Software is highly correlated on real data

We ran each software as described on real admixed individuals from the ASW (African Ancestry in Southwest US) population of the 1000 Genomes project with the YRI and CEU populations as reference panels. Local ancestry estimates were highly correlated between each software (Table 5). Additionally, to show the robustness of these estimates, we plot the mean local African ancestry estimated by each software against the first principal component of the genotypes, which is known to be an estimate of global African ancestry (Fig. 7) (Zhong, Perera & Gamazon, 2019). The local ancestry estimate was highly correlated with PC1 for all software tools (R2 > 0.960, Table 5), with no significant difference between tools (p > 0.621).

Discussion

Local ancestry estimation is a key step in adjusting for potential population stratification in admixed populations and in elucidating the effect of ancestry specific loci on complex traits. Given the wide variety of tools available to perform local ancestry estimation, it is necessary to explore how each performs in a particular context. Here, we focused on recent human admixture within African American and Hispanic populations, and performed complexity and accuracy analyses of five different software tools using simulated and real data.

Figure 7 Estimated African ancestry proportion in the ASW population (African Ancestry in Southwest US) is correlated with the first principal component.

We plot the the mean local ancestry proportion of African ancestries estimated by each software against the first principal component of genotypes, a known estimate of global African ancestry, to validate the robustness of local ancestry estimates. The local ancestry estimate was highly correlated with PC1 for all software tools (R2 > 0.96), with no significant difference between tools (p > 0.62).

We did not consider instances of ancient admixture despite ELAI and Loter reporting robust performance in such instances (Guan, 2014; Dias-Alves, Mairal & Blum, 2018). In addition, Loter was designed to be compatible with many different species and both Loter and ELAI may require more fine-tuning of software parameters beyond the default settings than the other methods, especially in cases of 3-way admixture.

Here we report on how memory and time usage scale with number of individuals and not SNPs, as it is more common to scale studies by population size than by genome size. However, it is expected that most if not all software will increase in both time and memory usage given an increased number of SNPs. We find consistently that RFMix and ELAI had high performance relative to other software, with RFMix beating ELAI marginally in several cases. While RFMix has a relatively low memory overhead, its runtime scales quadratically, severely limiting its scalability at standard GWAS sample sizes.

An important consideration in all cases is the availability of high quality reference data. Currently, Native American genetic data is not widely available due to historical misuse of DNA that have raised barriers between tribal communities and the genetics community (Garrison, 2013; TallBear, 2013). Here, individuals in the PEL population with estimated Native American ancestries greater than 77% were a proxy for non-admixed individuals of Native American descent. However, this introduced noise because these individuals in PEL contain significant continental admixture. This noise likely caused the HIS and 3WAY simulated populations to underperform compared to the AFA population. In spite of this, these simulations show robust performance of several software.

All the software are highly accessible for a user with root access. For individuals working without root access, RFMix, ELAI, and LAMP-LD are distributed with precompiled binaries and MOSAIC is distributed with a simple R interface that requires no compilation. Loter is simple to compile and install even without root access given that all prerequisites are installed on the machine. Each software differs in its flexibility and degree of user control. LAMP-LD and Loter are the easiest to run but have relatively few options and simple documentation. Loter also has the benefit of being the only software that directly accepts VCF files as input, though generating the inputs of the other software is fairly straightforward. Each accepts some variation of files that can be generated by the popular bcftools (Li, 2011) or PLINK (Chang et al., 2015). RFMix, MOSAIC, and ELAI each have a large number of customizable user options pertaining to their respective algorithms that the savvy user can use to optimize their analysis. Subsequently, the documentation of these three software are fairly detailed. MOSAIC is additionally written with several useful visualization functions for its output. Among these five software, only LAMP-LD does not preserve the phase data of its input genotypes.

Conclusion

We find that in cases of two-way simulated admixture, each software performs similarly well with RFMix having the highest median performance in all tested populations. While RFMix performs the best across methods, its scalability with regards to time may give weight to considering other software. However this issue can be circumvented by dividing haplotypes across multiple runs. Robust, scalable local ancestry estimation software are crucial for equitable implementation of genetics and genomics in medicine.

Supplemental Information

Supplemental Information 1 Distribution of global ancestries in simulated admixed AFA population

Each bar is an individual with their percentages of global ancestries represented per haplotype.

Click here for additional data file.

Supplemental Information 2 Distribution of global ancestries in simulated admixed HIS population

Each bar is an individual with their percentages of global ancestries represented per haplotype.

Click here for additional data file.

Supplemental Information 3 Distribution of global ancestries in simulated admixed 3WAY population

Each bar is an individual with their percentages of global ancestries represented per haplotype.

Click here for additional data file.

Supplemental Information 4 Pairwise software comparison in AFA

95% family confidence intervals for Tukey’s test when running each software on a two way AFA admixed population. We find significant differences between the following pairs: RFMix & Loter (p = 4.0e − 13), RFMix & ELAI (p = 3.7e − 7), RFMix & MOSAIC (p < 2.14e − 14), LAMP-LD & Loter (p = 1.0e − 9), LAMP-LD & ELAI (p = 1.0e − 4), LAMP-LD & MOSAIC (p < 2.14e − 14), ELAI & MOSAIC (p = 1.0e − 7), and Loter & MOSAIC (p = 2.3e − 3). All other pairs were found to not have significantly different means

Click here for additional data file.

Supplemental Information 5 Pairwise software comparison in HIS

95% family confidence intervals for Tukey’s test when running each software on a two way HIS admixed population. We find significant differences between the following pairs: RFMix & LAMP-LD (p = 8.0e − 9), RFMix & Loter (p < 2.14e − 14), RFMix & ELAI (p = 2.6e − 6), RFMix & MOSAIC (p < 2.14e − 14), ELAI & Loter (p < 2.14e − 14). ELAI & MOSAIC (p = 1.2e − 2). MOSAIC & Loter (p < 2.14e − 14), and LAMP-LD & Loter (p < 2.14e − 14). All other pairs were not found significantly different.

Click here for additional data file.

Supplemental Information 6 Pairwise software comparison in 3WAY

95% family confidence intervals for Tukey’s test when running each software on three way admixed population. We find significant differences between the following pairs: RFMix & LAMP-LD (p = 4.6e − 5), RFMix & Loter (p < 2.14e − 14), RFMix & MOSAIC (p < 2.14e − 14), RFMix & ELAI (p = 1.8e − 9), LAMP-LD & Loter (p < 2.14e − 14), LAMP-LD & MOSAIC (p = 1.2e − 12), ELAI & Loter (p < 2.14e − 14), ELAI & MOSAIC (p = 1.8e − 7), and MOSAIC & Loter (p < 2.14e − 14). All other pairs were not found significantly different.

Click here for additional data file.

We thank Dr. Catherine Putonti for her feedback during development of this project.

Additional Information and Declarations

Competing Interests

Author Contributions

Data Availability

The authors declare there are no competing interests.

Ryan Schubert and Angela Andaleon conceived and designed the experiments, performed the experiments, analyzed the data, prepared figures and/or tables, authored or reviewed drafts of the paper, and approved the final draft.

Heather E. Wheeler conceived and designed the experiments, authored or reviewed drafts of the paper, and approved the final draft.

The following information was supplied regarding data availability:

All genomic data used are publicly available from the 1000 Genomes Project: https://www.internationalgenome.org/.

Scripts used to format data, run software, and estimate accuracy can be found at Github: https://github.com/WheelerLab/LAI_benchmarking.

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
