# Peer review of "Comparing local ancestry inference models in populations of two- and three-way admixture"

_PeerJ, doi:10.7717/peerj.10090_

## Round 0.1 · original submission · Minor Revisions

Please see the reviewer's comments as follows. I encourage you to make edits accordingly and resubmit it again.

Reviewer 1 ·

Basic reporting

no comment

Experimental design

no comment

Validity of the findings

no comment

Additional comments

This is a well written paper which introduced and compared five major local ancestry inference models of two-way and three-way admixture populations.
I have no major concerns, but I do think that there are points need to be revised.

1. Line 93: For “u=0.77”, it would be better to specify the meaning of this statistic.
2. Line 90-94: The authors argued that the PEL population could serve as a reasonable proxy for Native American ancestral population in this study. I do agree that the PEL have more Native American ancestries than other admixed American populations in 1000G, however, I still feel cautious that PEL could be a good reference population for Native Americans. It would be good to find some other populations, but I understand it would be hard by considering of the scale of the data, data merging etc. I think another way to build a good reference panel would be estimating the Native American ancestry for each PEL individual, and then select the first 10% of individuals who has the most Native American ancestry.
3. Line 96 and Table 2: When simulating the admixed populations, the authors fixed the components of each ancestral populations. Please cite the reference papers to prove that these numbers are appropriate to be set.
4. Line 103: I do concern that chromosome 22 may not be a good example here, since it is short and may contain a lot of structural variations. Maybe simulating the first 1-10Mb of Chromosome 1 is a better way to do so.
5. For Figure 6, besides using the correlation coefficient, I would suggest using another statistic (y-axis) to measure the accuracy, e.g. the absolute difference of the estimated ancestries. It is also interesting to explore why the performances were bad in some cases for each software (low values of y-axis).

Reviewer 2 ·

Basic reporting

In this study, the performances of five methods for local ancestry estimation were compared: LAMP-LD (2012), RFMix (2013), ELAI (2014), Loter (2018), and MOSAIC (2019), with respect to accuracy, runtime, memory usage, and usability. Authors found that in cases of two-way admixture, each software performs similarly well with RFMix and ELAI having the highest median performance. In three-way admixture, RFMix performs best overall while its runtime scales quadratically.

Experimental design

no comment.

Validity of the findings

no comment.

Additional comments

1. Line 205: “We report the estimated β1, model R2, and ANOVA p-value for each combination of software and model.” However, R2 and ANOVA p-value of linear and quadratic models were reported in Table 4. Please check.
2. Figure 4, 5: It seems a caption is omitted. There is no need to repeat “We test to see if there is a significant difference ……” in the legend. In addition, were different methods (one-way ANOVA, two-way ANOVA) employed to test the differences as respect to the runtime and the maximum memory usage? Please check.
3. Figure 5 legend: Doesn’t it test the difference between the runtime of each software when increasing the number of ancestries estimated? However, in the legend, you say that it is between number of ancestries and the maximum memory usage.
4. Table 5: Removing the redundant data above the diagonal would make for a more presentable table.
5. Line 387: The reference of Loter is not correctly cited.

---

## Round 0.2 · accepted · Accept

Thanks for your re-submission. I believe the current manuscript is ready for publication. Congratulations!